# ICL-SAM: Synergizing In-context Learning Model and SAM in Medical Image Segmentation

**Jiesi Hu**[1,2]                                                405323011@QQ.COM
**Yang Shang**[1]                                               23S052012@STU.HIT.EDU.CN
**Yanwu Yang**[1,2]                                             20B952019@STU.HIT.EDU.CN
**Xutao Guo**[1,2]                                              18B952052@STU.HIT.EDU.CN
**Hanyang Peng**[2]                                             PHILOSO_PHY0922@163.COM
**Ting Ma**[*1,2,3,4]                                           TMA@HIT.EDU.CN

[1] *Electronic and Information Engineering School, Harbin Institute of Technology (Shenzhen), Shenzhen, China*

[2] *Peng Cheng Laboratory, Shenzhen, China*

[3] *Guangdong Provincial Key Laboratory of Aerospace Communication and Networking Technology, Harbin Institute of Technology (Shenzhen), Shenzhen, China*

[4] *International Research Institute for Artificial Intelligence, Harbin Institute of Technology (Shenzhen), Shenzhen, China*

**Editors:** Accepted for publication at MIDL 2024

## Abstract

Medical image segmentation, a field facing domain shifts due to diverse imaging modalities and biomedical domains, has made strides with the development of robust models. The In-Context Learning (ICL) model, like UniverSeg, demonstrates robustness to domain shifts with support image-label pairs in varied medical imaging segmentation tasks. However, its performance is still unsatisfied. On the other hand, the Segment Anything Model (SAM) stands out as a powerful universal segmentation model. In this work, we introduce a novel methodology, ICL-SAM, that integrates the superior performance of SAM with the ICL model to create more effective segmentation models within the in-context learning paradigm. Our approach employs SAM to refine segmentation results from ICL model and leverages ICL model to generate prompts for SAM, eliminating the need for manual prompt provision. Additionally, we introduce a semantic confidence map generation method into our framework to guide the prediction of both ICL model and SAM, thereby further enhancing segmentation accuracy. Our method has been extensively evaluated across multiple medical imaging contexts, including fundus, MRI, and CT images, spanning five datasets. The results demonstrate significant performance improvements, particularly in settings with few support pairs, where our method can achieve over a 10% increase in the absolute Dice coefficient compared to cutting edge ICL model. Our code will be publicly available.

**Keywords:** Segmentation, In-context learning, SAM

## 1. Introduction

Image segmentation represents a pivotal challenge in medical image analysis, and deep learning has increasingly become the predominant approach for this task ([Ronneberger

---

* Corresponding authors

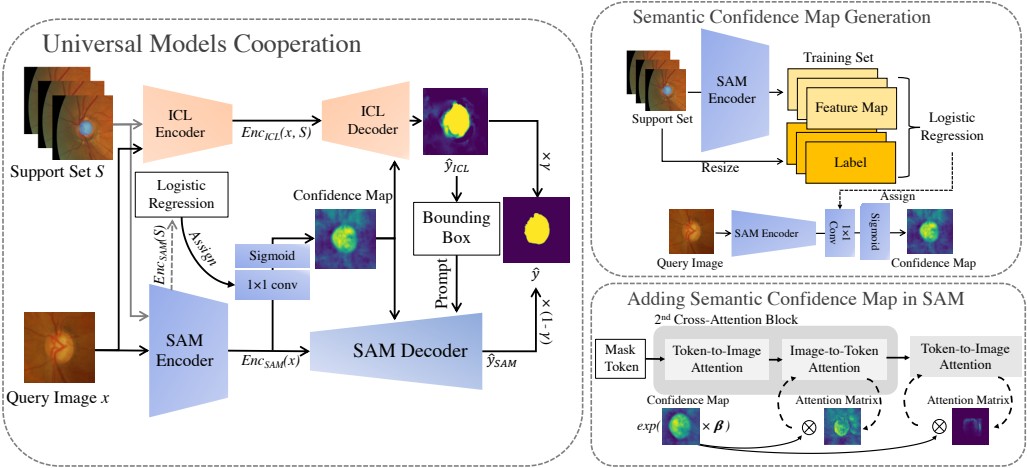

Figure 1: Illustration of the Universal models cooperation framework, Semantic confidence map generation module and Adding semantic confidence map in SAM module.

et al., 2015; Isensee et al., 2021; Chen et al., 2021b; Dolz et al., 2018). The field grapples with domain shift due to the heterogeneity of imaging modalities and the diversity within biomedical domains (Liu et al., 2023). While existing segmentation tools are technologically advanced, they are often restricted to specific tasks or related domains. This specialization limits their ability to address domain shifts and diverse scenarios, particularly in the context of evolving clinical and scientific requirements in medical imaging. Despite various fine-tuning techniques being proposed (Liu et al., 2023; Chen et al., 2021a; Yang et al., 2022; Hu et al., 2023), the necessity for extensive computational resources and specialized machine learning expertise poses significant barriers in actual applications.

Originating from the field of natural language processing, In-Context Learning (ICL) has recently emerged as a promising methodology for developing universal segmentation models robust to domain shifts. Models such as UniverSeg (Butoi et al., 2023) and Neuralizer (Czolbe and Dalca, 2023) applied this approach, constructing models that can adapt to new tasks or domains by leveraging the support set present in input data. This circumvents the need for extensive retraining and achieves promising results in few-shot scenarios. However, current state-of-the-art ICL models like UniverSeg(Butoi et al., 2023) still face challenges due to their suboptimal performance, particularly with limited support set data.

On the other hand, the Segment Anything Model (SAM) (Kirillov et al., 2023) represents another universal segmentation model capable of generating satisfactory segmentation masks with bounding box or point prompts. Since SAM's introduction, numerous studies have explored its application in image segmentation (Zhang et al., 2023). Huang et al. (Huang et al., 2023) assessed SAM's effectiveness across various medical datasets, noting that bounding box prompts generally outperform point prompts. Ma et al. (Ma and Wang, 2023) achieved promising results by fine-tuning SAM with a substantial annotated medical segmentation dataset. However, the usage paradigm of SAM differs from that of ICL models. SAM necessitates providing prompts for each query image, whereas ICL models require only a set of fixed image-label pairs.

In this paper, we propose a method called ICL-SAM that utilizes SAM to refine the suboptimal segmentation results produced by ICL model and allows ICL model to provide targeted prompts for SAM, thereby eliminating the necessity for manual prompt provision. Furthermore, we harness semantic confidence maps derived from SAM's feature maps, which are replete with rich semantic information, to guide the generation of results and, thus, improve accuracy. ICL-SAM has been comprehensively evaluated in various contexts, including fundus, MRI, and CT images. It achieves significant performance improvements over cutting edge ICL models, notably in few-shot scenarios, where it achieves an increase over 10% in the Dice coefficient. The key contributions of our work are summarized as follows:

- We introduce an innovative method that employs the powerful SAM model to bolster and refine in-context learning model in medical image segmentation.

- We propose a novel semantic confidence map generation technique and its integration within our framework to enhance segmentation performance, based on the in-context learning framework.

- Our methodology is extensively evaluated across three types of images and five datasets, demonstrating significant improvements in model performance, especially in situations with limited support sets.

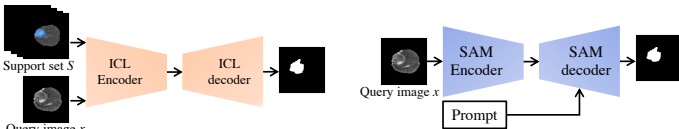

Figure 2: Workflows of ICL model and SAM during inferencing

## 2. Method

Consider the set $\{(x_i, y_i)\}_{i=1}^n$ of image-label pairs for a segmentation task. We utilize the advanced ICL model UniverSeg (Butoi et al., 2023) in our experiments. The UniverSeg model learns a function $\hat{y}_{\text{ICL}} = \text{Dec}_{\text{ICL}}(\text{Enc}_{\text{ICL}}(x, S))$, utilizing a CNN encoder and decoder architecture. It predicts a label map for an input $x$ in accordance with the task-specified support set $S = \{(x_j, y_j)\}_{j=1}^m$, which consists of available example image-label pairs, where $m$ denotes the number of pairs in the support set. For the SAM model, the prediction map is defined as $\hat{y}_{\text{SAM}} = \text{Dec}_{\text{SAM}}(\text{Enc}_{\text{SAM}}(x), \text{prompt})$. Figure 2 shows the workflow of these two models during inference.

### 2.1. Universal Models Cooperation

The architecture of our cooperative methodology is depicted in Figure 1. This module operates by iteratively generating bounding boxes from the segmentation outputs of the ICL model. These bounding boxes are then employed as prompts within the SAM framework, thereby achieving refined segmentation outcomes. Empirical studies, such as those by (Ma

and Wang, 2023; Huang et al., 2023), have substantiated the effectiveness of bounding boxes as potent prompts for medical image segmentation. In addition, to fully harness the capabilities of the SAM encoder, we generate a semantic confidence map from its feature map outputs, specifically designed to emphasize the foreground regions. This confidence map plays a pivotal role in concurrently enhancing the segmentation performance of both the ICL model and SAM.

The final output map fuses results from both SAM and ICL models, formulated by the following equation:

$$\hat{y} = \gamma \hat{y}_{\text{ICL}} + (1 - \gamma) \hat{y}_{\text{SAM}}, \tag{1}$$

where $\hat{y}$ denotes the prediction of our method. Considering that the performance of the ICL model is enhanced with an increase in support set size $m$, we dynamically adjust the $\gamma$ value to $\gamma = a(1 - e^{-\tau m})$ to increase the weight of ICL, where $\tau$ is a temperature factor.

## 2.2. Semantic Confidence Map Generation

Given that the support set comprises labeled data, it is feasible to employ the universal features extracted by SAM to train a Logistic Regression (LR) model. This model is adept at generating a semantic confidence map for the query image that accentuates the target region, as delineated in Figure 1. In the training phase, features $\text{Enc}_{\text{SAM}}(S) \in \mathbb{R}^{m \times 64 \times 64 \times 256}$ from the support set are utilized as training data, where $m$ denotes the size of the support set. These features are reshaped into $\text{Enc}_{\text{SAM}}(S) \in \mathbb{R}^{4096m \times 256}$ to serve as the input for the LR model, with the corresponding labels in the support set acting as the ground truth $Y \in \mathbb{R}^{4096m}$. The SAGA algorithm (Defazio et al., 2014), known for its efficiency with large training sets, alongside binary cross-entropy loss, is utilized for parameter optimization. After training, the semantic confidence map can be computed as follows:

$$C = \text{sigmoid}(\delta \odot \text{Enc}_{\text{SAM}}(x) + \delta_0), \tag{2}$$

where $\delta$ represents the parameters of the LR model. Furthermore, since the dot product in feature dimensions can be replaced by a convolution layer with a $1 \times 1$ kernel, the parameters of the trained logistic regression model are assigned to a convolution layer during actual inference. This approach enables the model to exploit GPU parallelism, thus ensuring inference speed. During inference, the confidence map for a query image $x$ is computed by $C = \text{sigmoid}(\text{conv}_{1 \times 1}(\text{Enc}_{\text{SAM}}(x)))$, where the confidence map is subsequently interpolated to $C \in \mathbb{R}^{h \times w}$, where $h$ and $w$ represent the spatial size of the target layer to which $C$ will be applied. Note that other pretrained models can be used to build confidence maps instead of SAM using this method.

## 2.3. Adding Semantic Confidence Map in SAM

The bounding boxes provided by the ICL model may lack precision. To alleviate this, we introduce explicit semantic guidance into the decoder of SAM, enhancing its focus on foreground regions. As depicted in Figure 1, we incorporate the generated confidence map into the second cross-attention block of the image-to-token attention and the final token-to-image attention blocks within SAM's decoder. While there are other attention blocks in SAM's decoder, we have found that changing just these two blocks suffices. Specifically, we

have modified the original attention matrices $I \in \mathbb{R}^{h \times w}$ corresponding to the mask token for these blocks as follows:

$$I_{ICL-SAM} = \mathrm{softmax}(I \odot e^{\beta\text{Z-Norm}(C)}), \tag{3}$$

where Z-Norm($C$) denotes the z-score normalization, and $\beta = 2$ serves as a balancing factor. A larger value of $\beta$ exerts greater influence of the confidence map on the original attention matrices. This modification allows SAM to more effectively concentrate on foreground features even when the bounding box is not accurate, thereby enhancing overall segmentation accuracy.

### 2.4. Adding Semantic Confidence Map in ICL Model

The spatial information provided by the semantic confidence map can be beneficial for the ICL model, particularly when its segmentation is imprecise. Therefore, we incorporate the confidence map as spatial attention into the decoder of the ICL model. Specifically, the process of adding the confidence map in UniverSeg is defined as follows:

$$\mathrm{Dec}^i_{\text{ICL-SAM}}(x) = \mathrm{Norm}\left(\mathrm{Dec}^i_{\text{ICL}}(x) \odot e^{\alpha\text{Z-Norm}(C)}\middle|\mathrm{Dec}^i_{\text{ICL}}(x)\right), \tag{4}$$

where $\mathrm{Dec}^i_{\text{ICL}}(x)$ represents the feature map at the $i$th decode layer of the UniverSeg model. The function Norm is employed to ensure the consistency of the L2 norm of the feature map, defined as $\mathrm{Norm}(B|A) = \frac{B||A||_2}{||B||_2}$. This attention mechanism is incorporated at each layer of the decoder in UniverSeg, analogous to the Attention U-Net (Oktay et al., 2018). The larger the value of $\alpha$, the more significant is the impact of the confidence map on the UniverSeg model. When $\alpha = 0$, the outcome is equivalent to the vanilla UniverSeg. We dynamically adjust the $\alpha$ value as $\alpha = be^{-\tau m}$. When the support set size is larger, we reduce the effect of the semantic confidence map.

### 2.5. Iterative Bounding Box Generation

We generate the bounding box prompts for SAM using the output pseudo-label of the ICL model. To eliminate noise within the prediction map, we first apply morphological shrinkage to the pseudo-label, reducing it to reduce the area of the pseudo-labels to 90% of their original size. This step helps eliminate some of the finer noise. Subsequently, we use morphological inflation to restore the retained pseudo-labels to their original size, ensuring the accuracy of the bounding boxes generated afterward. Subsequently, we iteratively select foreground components and generate bounding boxes around the component, which are then inputted as prompts into the SAM decoder. The final output map from the SAM decoder unifies the maps generated for each bounding box: $\mathrm{Dec}_{\text{SAM}}(\mathrm{Enc}_{\text{SAM}}(x)) = \bigcup_i \mathrm{Dec}_{\text{SAM}}(\mathrm{Enc}_{\text{SAM}}(x), \mathrm{prompt}_i)$. This method can effectively handle scenarios involving multiple separated targets within an image.

## 3. Experiments and Discussion

### 3.1. Datasets

Our methodology was evaluated across three distinct scenarios, encompassing the segmentation of fundus, brain MRI, and kidney CT images. Each dataset consists of designated meta support set and query sets. We randomly selected image-label pairs to constitute the support set from the meta support set and conducted inference on the query set.

(1) For the segmentation of the optic disc and cup in retinal fundus images, we utilized datasets from the **REFUGE** challenge (Orlando et al., 2020), **RIM-ONE-r3** (Fumero et al., 2011), and **Drishti-GS** (Sivaswamy et al., 2015). The composition of the meta support and query sets for these datasets was 320/80, 99/60, and 50/51 images, respectively.

(2) Whole tumor segmentation was performed on T1, T1ce, T2, and FLAIR modalities using the **BraTS2020** dataset (Bakas et al., 2018), focusing on low-grade glioma cases. The meta support set and the query set were randomly divided with 53 and 23 cases.

(3) The **Kits23** dataset (Heller et al., 2023) was utilized for combined segmentation of kidney and tumor, with the dataset being randomly partitioned into meta support and query subsets containing 245 and 244 cases, respectively.

For the 3D MRI and CT datasets, we extracted 2D slices that contained the segmentation targets. The preprocessing of images was aligned with the protocols in UniverSeg (Butoi et al., 2023) and SAM (Kirillov et al., 2023; Ma and Wang, 2023). Detailed description of the datasets could be found in appendix.

### 3.2. Implementation Details and Comparison Models

Our experiments were conducted on NVIDIA V100 GPUs equipped with 32GB of memory. For each inference scenario, we randomly selected support sets 10 times, calculating their mean results to derive the final outcome. The Dice coefficient, which quantifies the overlap between the predicted segmentation and the ground truth, was employed for evaluation. In these experiments, the parameters $a$, $b$, $\tau$ were set to 0.5, 0.3, and 0.1, respectively, values that were determined to be optimal through our testing. Note that our model does not require fine-tuning the parameters of the ICL and SAM models. Thus, ICL-SAM can be directly applied to other tasks without retraining.

To ascertain the efficacy of our approach, comparisons were made with both UniverSeg(Butoi et al., 2023) and Neuralizer(Czolbe and Dalca, 2023), which are state-of-the-art in-context learning models in the realm of medical imaging. UniverSeg is a universal segmentation model, and Neuralizer, trained on neuroimaging data, is versatile in performing a variety of tasks beyond segmentation. Regarding the SAM model, we evaluated both the original SAM(Kirillov et al., 2023) checkpoint and the MedSAM(Ma and Wang, 2023) checkpoint of the ViT-B model.

### 3.3. Comparison Results

Table 1 displays the Dice score of our model across various support set sizes and datasets. In the case of the Fundus dataset, we present the mean segmentation values for the optic disc and cup across three datasets. For the BraTs2020 dataset, the average segmentation outcomes across all modalities are shown. For the Kits23 dataset, we illustrate the results

Table 1: Performance comparison of different models on multiple datasets.

| Dataset | Model | Support set size | | | | | | |
|---|---|---|---|---|---|---|---|---|
| | | **1** | **2** | **4** | **8** | **16** | **32** | **64** |
| Fundus | UniverSeg | 0.5797 | 0.7081 | 0.7529 | 0.7896 | 0.8148 | 0.8290 | 0.8344 |
| | Neuralizer | 0.6774 | 0.7109 | 0.7314 | 0.7325 | 0.7531 | 0.7515 | 0.7512 |
| | UniverSeg+SAM | 0.6909 | 0.7347 | 0.7716 | 0.8030 | 0.8173 | 0.8293 | 0.8339 |
| | UniverSeg+MedSAM | **0.7391** | **0.7889** | **0.8057** | **0.8264** | **0.8391** | **0.8467** | **0.8499** |
| | SAM+GT | | | | 0.7161 | | | |
| | MedSAM+GT | | | | 0.8873 | | | |
| BraTs | UniverSeg | 0.2078 | 0.3059 | 0.4706 | 0.5780 | 0.6704 | 0.7277 | 0.7747 |
| | Neuralizer | 0.2161 | 0.2284 | 0.2540 | 0.2563 | 0.3203 | 0.3760 | 0.4750 |
| | UniverSeg+SAM | 0.2874 | 0.3993 | 0.5269 | 0.6184 | 0.6857 | 0.7454 | 0.7799 |
| | UniverSeg+MedSAM | **0.3387** | **0.4584** | **0.5879** | **0.6579** | **0.7191** | **0.7586** | **0.7899** |
| | SAM+GT | | | | 0.8398 | | | |
| | MedSAM+GT | | | | 0.8447 | | | |
| Kits23 | UniverSeg | 0.4487 | 0.5671 | 0.7179 | 0.7838 | 0.8340 | 0.8500 | 0.8646 |
| | Neuralizer | 0.3715 | 0.5278 | 0.5510 | 0.6384 | 0.6515 | 0.6694 | 0.6728 |
| | UniverSeg+SAM | **0.6455** | **0.7182** | **0.8065** | **0.8439** | **0.8629** | **0.8764** | **0.8843** |
| | UniverSeg+MedSAM | 0.6272 | 0.6493 | 0.7438 | 0.8053 | 0.8504 | 0.8633 | 0.8757 |
| | SAM+GT | | | | 0.9510 | | | |
| | MedSAM+GT | | | | 0.9371 | | | |

of the combined segmentation of the kidney and tumor. Detailed results are available in the appendix. The SAM+GT demonstrates the performance of using ground truth map to generate bounding box input into SAM, which demonstrates the upper bound of the corresponding SAM model. In the appendix, we also present the Average Symmetric Surface Distance (ASSD) of our model.

A notable enhancement in performance is evident when the support set size is small. Employing our methodology with a support set size of 1 yields improvements of 15.94%, 13.09%, and 19.68% in the Dice coefficient across the three datasets, respectively. Furthermore, our approach consistently outperforms across all support set sizes and datasets. The integration of SAM and MedSAM demonstrates differential impacts in various scenarios. MedSAM is particularly advantageous for fundus and brain MRI, and the addition of SAM is more effective for CT kidney segmentation. This disparity is likely due to SAM's limited specialization in the medical field, leading to suboptimal performance in cases with less distinct boundaries like Fundus and BraTs, and superior results in kidney segmentation where boundaries are more obvious. For the Fundus data set, although the performance of SAM+GT is poor, UniverSeg+SAM can still bring improvements. We attribute this is to the benefits brought by the proposed confidence map. Although the current efficacy is yet to match that of MedSAM+GT, our method has successfully narrowed this gap, showcasing potential within the in-context learning framework.

Figure 3 presents our results in the form of a curve graph, where the shaded areas represent the standard deviation of the corresponding performances. Notably, for the Fundus and Kits23 datasets, our method with only 16 support instances achieves performance comparable to that of UniverSeg with a support size of 64. In the context of the BraTs2020 dataset, a support size of 32 achieves performance equivalent to UniverSeg with 64 support

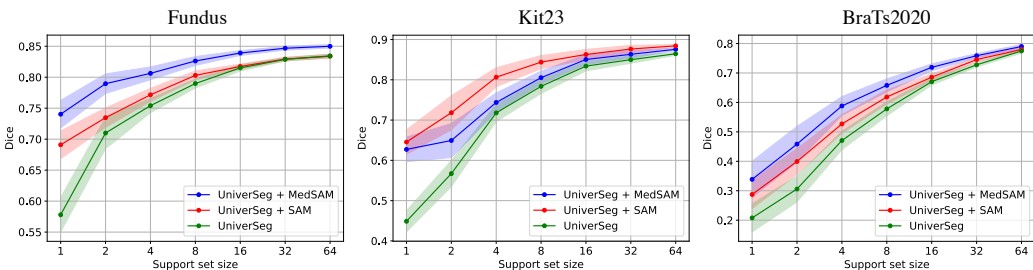

Figure 3: Comparison of models under different support sizes.

instances. This underscores the potential of our approach to significantly reduce the annotation burden on clinicians for in-context inference while maintaining robust performance.

Figure 4 demonstrates the enhancement process of UniverSeg through our methodology. It shows that UniverSeg's segmentation can be incomplete and coarse. The incorporation of the semantic confidence map leads to better segmentation outcomes. By processing the bounding boxes through SAM, we achieve a more comprehensive and accurate segmentation.

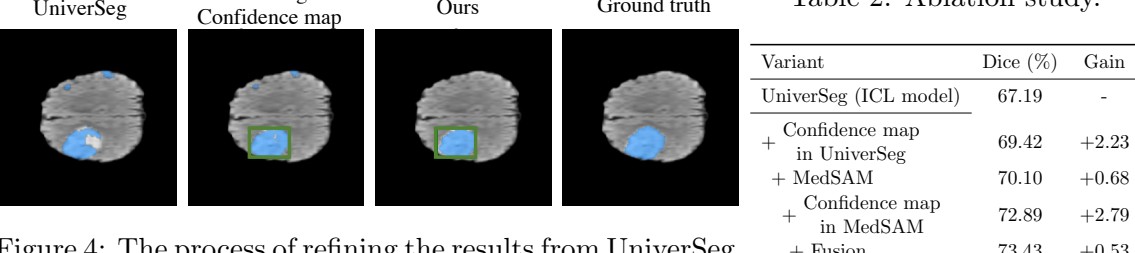

Figure 4: The process of refining the results from UniverSeg.

Table 2: Ablation study.

| Variant | Dice (%) | Gain |
|---|---|---|
| UniverSeg (ICL model) | 67.19 | - |
| + Confidence map in UniverSeg | 69.42 | +2.23 |
| + MedSAM | 70.10 | +0.68 |
| + Confidence map in MedSAM | 72.89 | +2.79 |
| + Fusion | 73.43 | +0.53 |

### 3.4. Ablation Study

Table 2 presents an ablation study, showcasing the mean performance across all three datasets and for every support set size. The task-specific ablation analyses are in the appendix. It is evident that incorporating SAM enhances the UniverSeg model's performance. Furthermore, the addition of semantic confidence into either UniverSeg or SAM markedly elevates the overall model efficacy. The Fusion approach, which combines the predictive outcomes of both the ICL and SAM models, also demonstrates improvement, particularly when the support set is large, because, in such scenarios, UniverSeg is capable of providing an relatively accurate segmentation mask.

## 4. Conclusion

Our proposed methodology capitalizes on SAM's high precision in segmentation and the ICL model's ability to provide contextual support. The integration of a semantic confidence map further enhances segmentation accuracy. Our comprehensive evaluations demonstrate the effectiveness of the proposed framework, particularly in scenarios with limited support sets. Additionally, this framework reduces the need for extensive manual input for SAM. Our research highlights the significant potential of the in-context learning paradigm, suggesting opportunities for future enhancements.

## Acknowledgments

This work was supported in part by grants from the National Natural Science Foundation of P.R. China (62276081), and The Major Key Project of PCL (PCL2021A06)

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

## Appendix A. Dataset Details

The Table 3 provides a comprehensive summary of the datasets utilized in the study, detailing their modality, object of interest, and the distribution of images across the meta support and query sets.

## Appendix B. Experiment Details

The Tables 4 and 5 showcase the detailed performance outcomes of various models across different support set sizes and datasets, specifically focusing on medical image segmentation tasks like Fundus (Disc and Cup) and BraTs2020 (FLAIR, T1, T1CE, T2), as well as the Kits23 dataset. Each model, including Neuralizer, UniverSeg, are evaluated for their segmentation efficacy as indicated by the support set sizes ranging from 1 to 64, consistent with the naming in 1. Additionally, we present ablation study data for different tasks to provide readers with a clearer understanding of our method's performance in various scenarios. The naming convention for the ablation study section is in line with that used in Tables 2, and '+ Fusion' is actually the same model as 'MedSAM+UniverSeg'.

A notable observation is the consistent performance improvement when confidence maps are integrated into both UniverSeg and MedSAM models, illustrating the value of semantic information in refining segmentation results. The Fusion approach, which combines the strengths of UniverSeg and MedSAM, generally yields the highest performance across most datasets and support sizes, underscoring the effectiveness of leveraging multiple models' capabilities in concert. Moreover, the comparison with SAM+UniverSeg and ground truth-enhanced versions (SAM+GT and MedSAM+GT) provides a benchmark, showing the potential ceiling of segmentation performance with these methodologies.

Overall, these results highlight the potential of advanced in-context learning models and their combinations to address the challenges of medical image segmentation, especially in scenarios with limited labeled data. The consistent performance gains across different datasets and support set sizes underscore the robustness and adaptability of the proposed methodologies.

To provide a comprehensive demonstration of our model's performance, we also include the Average Symmetric Surface Distance (ASSD) in Table 6. ASSD measures the mean distance between corresponding points on the surfaces of two objects, and is widely used in fundus and tumor segmentation. For consistency in comparison, all images were resized to $512 \times 512$ when computing the ASSD. The ASSD is reported in pixels. As indicated in the table, the results largely align with those in Table 1. UniverSeg+MedSAM achieves superior performance for the fundus and BraTS2020 datasets, while UniverSeg+SAM demonstrates

| Dataset | Modality | Object | Meta support set | | Query set | |
|---------|----------|--------|-------------------|-----------------|------------------|-----------------|
| | | | # 3D images | # 2D images | # 3D images | # 2D images |
| REFUGE | RGB | Fundus | - | 320 | - | 80 |
| RIM-ONE-r3 | RGB | Fundus | - | 99 | - | 60 |
| Drishti-GS | RGB | Fundus | - | 20 | - | 51 |
| BraTs2020 | T1, T1ce, T2, FLAIR | Brain | 53×4 | 3439×4 | 23×4 | 1487×4 |
| Kits23 | CT | Kidney | 245 | 6396 | 244 | 5630 |

Table 3: Summary of datasets used in the study.

the best outcomes for the Kits23 dataset. It is also noted that model improvements are more pronounced at smaller context sizes, with performance gains diminishing as context size increases. Nonetheless, even at a context size of 64, a significant improvement is observed for the Kits23 dataset, highlighting the exceptional performance of our method. Furthermore, there is a gap between our approach and MedSAM+GT, suggesting potential for further enhancement of in-context learning models.

Figure 4 presents additional visualization results, primarily illustrating how our outcomes evolve from the original ICL model (UniverSeg). The evolution process adheres to the naming convention of the ablation study in Tables 2. To distinctly showcase the effects, we mainly selected smaller context sizes (2 for the fundus dataset and 4 for others). From the figure, it is evident that incorporating the confidence map into the ICL model or integrating SAM significantly enhances the results. This also reveals that the main reason for the improvement offered by our method is the utilization of the powerful SAM to rectify inaccuracies produced by the ICL model. Moreover, our approach tends to yield good results when the segmentation from UniverSeg provides an approximate location of the target and generates precise bounding boxes. However, cases where UniverSeg's segmentation map misses the target result in SAM's inability to produce accurate results as well. This limitation is a primary factor in the persisting performance gap between our method and MedSAM+GT.

Figure 6 displays the hyperparameter sensitivity analysis. We assessed the effects of different hyperparameters on fundus datasets and found that optimal performance is attained when $a$, $b$, $\tau$, and $\beta$ are set to 0.5, 0.1, and 2.0, respectively. Given their effectiveness on the fundus datasets, these hyperparameters were also applied to other datasets, demonstrating the robustness and generalizability of our algorithm.

Table 4: Performance complarison across varying support set sizes and datasets, Part 1.

| Dataset | Model | Support set size | | | | | | |
|---|---|---|---|---|---|---|---|---|
| | | **1** | **2** | **4** | **8** | **16** | **32** | **64** |
| Fundus: Disc | Neuralizer | 0.8012 | 0.8514 | 0.8629 | 0.8716 | 0.8804 | 0.8851 | 0.8867 |
| | UniverSeg | 0.6611 | 0.8163 | 0.8648 | 0.8937 | 0.9117 | 0.9203 | 0.9235 |
| | SAM+UniverSeg | 0.8291 | 0.8664 | 0.8911 | 0.9066 | 0.9173 | 0.9240 | 0.9260 |
| | MedSAM+UniverSeg | 0.8716 | 0.9060 | 0.9157 | 0.9266 | 0.9331 | 0.9355 | 0.9373 |
| | + Confidence map in UniverSeg | 0.7330 | 0.8441 | 0.8756 | 0.8999 | 0.9146 | 0.9208 | 0.9236 |
| | + MedSAM | 0.8525 | 0.8862 | 0.8931 | 0.9055 | 0.9216 | 0.9227 | 0.9261 |
| | + Confidence map in MedSAM | 0.8710 | 0.9049 | 0.9151 | 0.9262 | 0.9319 | 0.9344 | 0.9349 |
| | + Fusion | 0.8716 | 0.9060 | 0.9157 | 0.9266 | 0.9331 | 0.9355 | 0.9373 |
| | SAM + GT | | | | 0.7706 | | | |
| | MedSAM + GT | | | | 0.9441 | | | |
| Fundus: Cup | Neuralizer | 0.5537 | 0.5704 | 0.5999 | 0.5935 | 0.6259 | 0.6180 | 0.6157 |
| | UniverSeg | 0.4983 | 0.5998 | 0.6410 | 0.6856 | 0.7179 | 0.7377 | 0.7453 |
| | SAM+UniverSeg | 0.5528 | 0.6029 | 0.6521 | 0.6994 | 0.7173 | 0.7346 | 0.7418 |
| | MedSAM+UniverSeg | 0.6066 | 0.6719 | 0.6956 | 0.7263 | 0.7451 | 0.7579 | 0.7625 |
| | + Confidence map in UniverSeg | 0.5222 | 0.6095 | 0.6455 | 0.6905 | 0.7214 | 0.7384 | 0.7455 |
| | + MedSAM | 0.5630 | 0.6257 | 0.6454 | 0.6715 | 0.6936 | 0.7131 | 0.7226 |
| | + Confidence map in MedSAM | 0.6060 | 0.6710 | 0.6968 | 0.7288 | 0.7455 | 0.7568 | 0.7622 |
| | + Fusion | 0.6066 | 0.6719 | 0.6956 | 0.7263 | 0.7451 | 0.7579 | 0.7625 |
| | SAM + GT | | | | 0.6615 | | | |
| | MedSAM + GT | | | | 0.8304 | | | |
| BraTs: FLAIR | Neuralizer | 0.2252 | 0.2519 | 0.2944 | 0.3026 | 0.3882 | 0.4460 | 0.5467 |
| | UniverSeg | 0.2566 | 0.3880 | 0.6193 | 0.7292 | 0.8024 | 0.8402 | 0.8668 |
| | SAM+UniverSeg | 0.3342 | 0.5095 | 0.6765 | 0.7810 | 0.8279 | 0.8598 | 0.8791 |
| | MedSAM+UniverSeg | 0.4299 | 0.5776 | 0.7430 | 0.8076 | 0.8502 | 0.8715 | 0.8874 |
| | + Confidence map in UniverSeg | 0.3311 | 0.4825 | 0.6547 | 0.7480 | 0.8091 | 0.8417 | 0.8669 |
| | + MedSAM | 0.3407 | 0.4974 | 0.7016 | 0.7512 | 0.8060 | 0.8214 | 0.8408 |
| | + Confidence map in MedSAM | 0.4337 | 0.5819 | 0.7462 | 0.8093 | 0.8479 | 0.8631 | 0.8729 |
| | + Fusion | 0.4299 | 0.5776 | 0.7430 | 0.8076 | 0.8502 | 0.8715 | 0.8874 |
| | SAM + GT | | | | 0.8743 | | | |
| | MedSAM + GT | | | | 0.8905 | | | |
| BraTs: T1 | Neuralizer | 0.2125 | 0.2181 | 0.2458 | 0.2478 | 0.2989 | 0.3592 | 0.4640 |
| | UniverSeg | 0.1736 | 0.2368 | 0.3735 | 0.4719 | 0.5687 | 0.6415 | 0.7032 |
| | SAM+UniverSeg | 0.2549 | 0.3257 | 0.4255 | 0.5054 | 0.5750 | 0.6545 | 0.7051 |
| | MedSAM+UniverSeg | 0.2819 | 0.3780 | 0.4872 | 0.5580 | 0.6199 | 0.6730 | 0.7171 |
| | + Confidence map in UniverSeg | 0.2314 | 0.3080 | 0.4174 | 0.4977 | 0.5785 | 0.6439 | 0.7033 |
| | + MedSAM | 0.2596 | 0.3537 | 0.3675 | 0.5223 | 0.5825 | 0.6338 | 0.6660 |
| | + Confidence map in MedSAM | 0.2846 | 0.3831 | 0.4928 | 0.5647 | 0.6205 | 0.6633 | 0.6980 |
| | + Fusion | 0.2819 | 0.3780 | 0.4872 | 0.5580 | 0.6199 | 0.6730 | 0.7171 |
| | SAM + GT | | | | 0.8192 | | | |
| | MedSAM + GT | | | | 0.8125 | | | |

Table 5: Performance complarison across varying support set sizes and datasets, Part 2.

| Dataset | Model | Support set size | | | | | | |
|---|---|---|---|---|---|---|---|---|
| | | **1** | **2** | **4** | **8** | **16** | **32** | **64** |
| BraTs: T1CE | Neuralizer | 0.2059 | 0.2014 | 0.2182 | 0.2201 | 0.2696 | 0.3058 | 0.3997 |
| | UniverSeg | 0.1907 | 0.2498 | 0.3557 | 0.4448 | 0.5585 | 0.6342 | 0.6951 |
| | SAM+UniverSeg | 0.2667 | 0.3242 | 0.4019 | 0.4884 | 0.5678 | 0.6546 | 0.6985 |
| | MedSAM+UniverSeg | 0.2850 | 0.3767 | 0.4675 | 0.5248 | 0.6087 | 0.6680 | 0.7092 |
| | + Confidence map in UniverSeg | 0.2394 | 0.3163 | 0.4041 | 0.4733 | 0.5714 | 0.6378 | 0.6953 |
| | + MedSAM | 0.2570 | 0.3394 | 0.4434 | 0.4912 | 0.5883 | 0.6398 | 0.6615 |
| | + Confidence map in MedSAM | 0.2874 | 0.3808 | 0.4737 | 0.5322 | 0.6102 | 0.6613 | 0.6925 |
| | + Fusion | 0.2850 | 0.3767 | 0.4675 | 0.5248 | 0.6087 | 0.6680 | 0.7092 |
| | SAM + GT | | | | 0.8007 | | | |
| | MedSAM + GT | | | | 0.8228 | | | |
| BraTs: T2 | Neuralizer | 0.2207 | 0.2422 | 0.2578 | 0.2546 | 0.3245 | 0.3930 | 0.4896 |
| | UniverSeg | 0.2104 | 0.3489 | 0.5337 | 0.6659 | 0.7521 | 0.7950 | 0.8336 |
| | SAM+UniverSeg | 0.2937 | 0.4378 | 0.6038 | 0.6989 | 0.7721 | 0.8126 | 0.8368 |
| | MedSAM+UniverSeg | 0.3579 | 0.5013 | 0.6538 | 0.7412 | 0.7975 | 0.8219 | 0.8461 |
| | + Confidence map in UniverSeg | 0.2855 | 0.4326 | 0.5763 | 0.6834 | 0.7592 | 0.7971 | 0.8336 |
| | + MedSAM | 0.3433 | 0.4300 | 0.6056 | 0.6689 | 0.7377 | 0.7650 | 0.7897 |
| | + Confidence map in MedSAM | 0.3603 | 0.5032 | 0.6574 | 0.7465 | 0.7967 | 0.8161 | 0.8325 |
| | + Fusion | 0.3579 | 0.5013 | 0.6538 | 0.7412 | 0.7975 | 0.8219 | 0.8461 |
| | SAM + GT | | | | 0.8649 | | | |
| | MedSAM + GT | | | | 0.8528 | | | |
| Kits23 | Neuralizer | 0.3715 | 0.5278 | 0.5510 | 0.6384 | 0.6515 | 0.6694 | 0.6728 |
| | UniverSeg | 0.4487 | 0.5671 | 0.7179 | 0.7838 | 0.8340 | 0.8500 | 0.8646 |
| | SAM+UniverSeg | 0.6455 | 0.7182 | 0.8065 | 0.8439 | 0.8629 | 0.8764 | 0.8843 |
| | MedSAM+UniverSeg | 0.6272 | 0.6493 | 0.7438 | 0.8053 | 0.8504 | 0.8633 | 0.8757 |
| | + Confidence map in UniverSeg | 0.5287 | 0.6143 | 0.7410 | 0.7940 | 0.8376 | 0.8510 | 0.8647 |
| | + MedSAM | 0.5864 | 0.6349 | 0.7320 | 0.7934 | 0.8296 | 0.8460 | 0.8515 |
| | + Confidence map in MedSAM | 0.6269 | 0.6450 | 0.7317 | 0.7866 | 0.8228 | 0.8402 | 0.8518 |
| | + Fusion | 0.6272 | 0.6493 | 0.7438 | 0.8053 | 0.8504 | 0.8633 | 0.8757 |
| | SAM + GT | | | | 0.9510 | | | |
| | MedSAM + GT | | | | 0.9371 | | | |

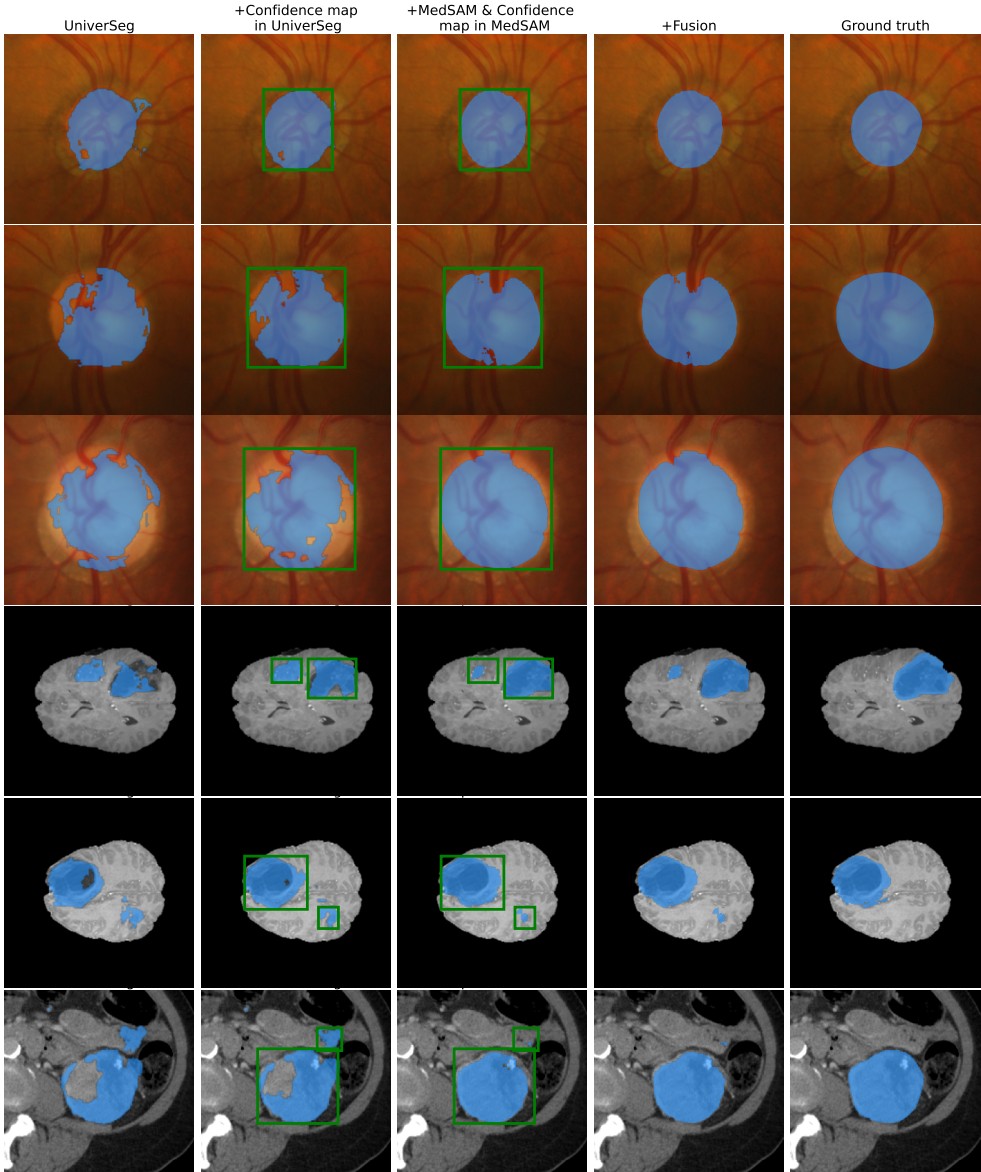

Figure 5: Details of refining the results from UniverSeg.

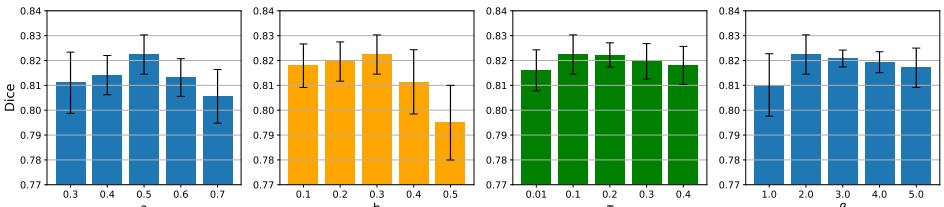

Figure 6: Sensitivity analysis of hyperparameters.

Table 6: Performance comparison of ASSD (pixel) across multiple datasets.

| Dataset | Model | Support set size | | | | | | |
|---------|-------|------|------|------|------|------|------|------|
| | | 1 | 2 | 4 | 8 | 16 | 32 | 64 |
| Fundus | UniverSeg | 23.18 | 17.83 | 14.45 | 12.44 | 11.13 | 10.41 | 10.17 |
| | Neuralizer | 22.38 | 19.43 | 16.03 | 14.65 | 13.97 | 13.64 | 13.36 |
| | UniverSeg+SAM | 17.78 | 15.80 | 13.80 | 12.23 | 11.02 | 10.52 | 10.02 |
| | UniverSeg+MedSAM | **16.99** | **13.41** | **10.86** | **10.01** | **9.50** | **8.91** | **8.71** |
| | SAM+GT | | | | 16.24 | | | |
| | MedSAM+GT | | | | 6.48 | | | |
| BraTs | UniverSeg | 46.12 | 41.55 | 36.18 | 26.59 | 20.74 | 17.88 | 14.23 |
| | Neuralizer | 63.80 | 52.92 | 51.75 | 46.76 | 41.71 | 33.71 | 27.04 |
| | UniverSeg+SAM | 48.25 | 39.37 | 34.35 | 26.61 | 20.27 | 17.48 | 14.32 |
| | UniverSeg+MedSAM | **39.74** | **35.10** | **30.32** | **23.06** | **18.64** | **16.29** | **13.34** |
| | SAM+GT | | | | 6.74 | | | |
| | MedSAM+GT | | | | 6.58 | | | |
| Kits23 | UniverSeg | 39.62 | 32.80 | 27.52 | 22.46 | 22.99 | 19.71 | 18.08 |
| | Neuralizer | 61.69 | 43.80 | 37.29 | 35.53 | 31.79 | 29.59 | 28.84 |
| | UniverSeg+SAM | **34.25** | **27.00** | **19.80** | **16.26** | **18.02** | **15.50** | **14.45** |
| | UniverSeg+MedSAM | 34.57 | 29.47 | 22.41 | 19.08 | 19.35 | 16.14 | 14.73 |
| | SAM+GT | | | | 4.19 | | | |
| | MedSAM+GT | | | | 5.47 | | | |

