# OpenReview forum: "ICL-SAM: Synergizing In-context Learning Model and SAM in Medical Image Segmentation"
_MIDL.io/2024/Conference — MIDL 2024 Oral_

### Official Review · Reviewer_nrDh · 2024-02-17

**Confidence:** 5
**Preliminary Rating:** 4
**Recommendation:** Poster

**Summary:**

This paper provided an in-context learning model called ICL-SAM, which is based on UniverSeg and SAM. The main objective of this model is to enhance the performance of in-context segmentation compared to UniverSeg. The authors have also introduced a novel technique for generating a semantic confidence map and integrated it into the combined framework. The experiments conducted on three different datasets demonstrate the effectiveness of the proposed methods. Overall, this paper provides valuable insights into improving in-context segmentation.

**Strengths:**

1. This paper has with clear objective in the set up setting, improving the performance of in-context learning. Therefore, the authors proposed reasonable methods and effective methods.
2. The paper is well-written and thus easy to follow.
3. The extensive and reasonable comparison and ablation experiments seem promising and prove the effectiveness of the method.

**Weaknesses:**

1. In the background, the authors claim SAM requires prompts and thus can be labor-intensive. Indeed this is true, but the proposed in-context model requires some label samples which might be more labor-intensive. Therefore, I recommend the authors reconsider their focus on the problem and it is better not to point out that SAM is “labor-intensive and time-consuming”.
2. The authors should pay more attention to the figure. For example, in Fig. 1, “token-to-image” is not unified because of letter capitalization, and the patch in the figure of “Confidence map” seems not unified with the “Attention matrix”. It might be better to check again.
3. In section 2.5, the sentence “the union of the maps produced by each bounding box” might mean that the results output unifies the maps with the three boxes, but it might be obscure from the description. In section 3.3, it is better to demonstrate the purpose of presenting the result of SAM+GT, which might indicate the upper bound of the setting.
4. In the datasets, the authors claim they conducted inference on the test set, but they do not describe clearly where they got the support set and query in the training stage and test stage.
5. In the implementation, I wonder whether the support sets are the same for all the comparison methods since different images in the support sets might bring unstable factors to the evaluation.
6. In Table 2, the authors might need to reorganize, since it might cause misunderstanding. For example, “Confidence map in MedSAM”  means MedSAM with confidence map, and “Fusion” should be explained before being abbreviated.

**Detailed Comments:**

See the section on weaknesses for more details.

**Justification Of The Preliminary Rating:**

This paper is well-written and provides enough experiments to evaluate the method that presents the improvement to the in-context segmentation. Although some details are not fully provided, I think this paper could be accepted after revising the existing issues stated in the weakness.

**Questions To Address In The Rebuttal:**

I recommend the authors provide more details and revise some of the referring arguments as stated in the section on weakness to improve the quality of the paper.

**Special Issue:**

Yes

---

> ### Author Response · Authors · 2024-03-15
>
> Thank you for your comments. I will address your points in the order of weaknesses mentioned:
>
> 1.	We highly value your suggestion concerning the labor-intensive and time-consuming aspects of our methodology. We have taken your advice into consideration and have made revisions to the relevant sections (introduction & abstract) in our paper, thereby redirecting our focus towards enhancing the efficacy of the in-context learning model.
>
> 2.	Thank you for your meticulous observation on Figure 1. We have addressed the capitalization issue in Figure 1 and rectified some minor problems as well.
>
> 3.	We have provided additional clarification in Sections 2.5 and 3.3 as per your suggestion. In short, the final output map from the SAM decoder unifies the maps generated for each bounding box. This method can effectively handle scenarios involving multiple separated targets within an image. In addition, the SAM+GT demonstrates the performance of using the ground truth map to generate bounding box input into SAM, which demonstrates the upper bound of the corresponding SAM model.
>
> 4.	Our description regarding the source of the support set and query set was indeed unclear. Consequently, we have renamed the training set and test set to meta support set and query set, respectively. Our support set is randomly selected from the meta support set, and evaluation is conducted on the images within the query set. Moreover, we did not fine-tune the weights of the SAM and ICL models. Therefore, we do not have a traditional "training stage," and our models can be directly applied to new tasks without the need for fine-tuning, with all evaluations occurring during the "test stage." We have emphasized this point in Section 3.2.
>
> 5.	You are correct that variations in the support set can influence the outcomes, particularly when the support set size is small. To address this concern in our experiments, we ensured that all comparison methods utilized the same support sets. Furthermore, to mitigate instability caused by the selection of support sets, we randomly drew 10 sets of support sets and calculated the average.
>
> 6.	We have reorganized Table 2, dividing each task into three sections: methods comparison, ablation study, and GT+SAM. We have also clarified the descriptions of Table 2 in the appendix for better understanding.
>
> We truly appreciate the time and effort you invested in reviewing our manuscript and offering insightful comments.

---

### Official Review · Reviewer_n42d · 2024-02-28

**Confidence:** 3
**Preliminary Rating:** 4
**Recommendation:** Poster
**Final Rating:** 4

**Summary:**

The authors show that their method, in context learning (the authors use UniverSeg and the semantic segmentation part of Neuralizer) combined with a foundation model (here: SAM and MedSAM) outperforms ICL-only methods. The authors test their method on three independent datasets and could show that SAM+ICL constantly outperforms ICL-only methods. Using an ablation study, the authors could show which elements of their method quantitatively contribute to the presented result.

**Strengths:**

The authors nicely show and explain the ICL method and its relation to SAM. The explanation of the method and the convenient use of the simple Logistic Regression and its use as 1x1 Conv to utilize GPUs is elegant. The analysis of foundation models in the field of medical imaging is important and current zeitgeist. The systematic analysis of the support set size in the ICL model is relevant.

**Weaknesses:**

From the text and table 1, I did not fully get the SAM+GT comparison. I understood that the bounding box was provided by the GT, and then the Dice score was computed with the GT itself. In Table 1, these values constantly outperform the ICL+SAM models. Does this mean, that accurate bounding boxes with the manual (?) prompt of SAM would create high-quality semantic segmentations?

As the authors claim that there is no need for manual prompts as in SAMs and the prompts are then provided by the ICL model, I would be interested in the prompts. How do they differ from the manual ones, what is the "salt" of the prompts allowing the performance gains? On the other hand, I thought SAMs do not need necessarily prompts but rather bounding boxes with a generic prompt. If the authors could give examples and show (maybe in the appendix) those, I would be very happy and intrigued.

**Detailed Comments:**

I have not seen a justification for the temperature coefficients/tau mentioned in the text. How did the authors chose these hyperparameters?

**Justification Of Final Rating:**

I am happy with the response of the authors and my rating stays the same as in the preliminary assessment. This is a nice paper, and especially with the amendments that the authors provided can be a very interesting story to be presented at MIDL.

**Justification Of The Preliminary Rating:**

The approach to combine in context learning and foundation models (SAM) is interesting for the community. The paper is well written and the approach is clear, the analysis is solid (on three datasets). Despite small weaknesses, I believe this paper adds knowledge to medical imaging with deep learning.

**Questions To Address In The Rebuttal:**

See Weaknesses.

**Special Issue:**

No

---

> ### Author Response · Authors · 2024-03-15
>
> Thank you for your suggestions, and I will respond to your points in order, from weaknesses to detailed comments:
> Weaknesses:
>
> 1.	We are grateful for your insightful inquiry regarding SAM+GT. Your understanding is correct. The term "SAM+GT" refers to the scenario where we use the ground truth to generate perfect bounding boxes to provide to SAM, and then calculate the Dice coefficient for SAM's output in this context. Indeed, providing accurate bounding box prompts does result in better segmentation from SAM. However, this outcome is expected since the bounding box prompt effectively offers strong supervisory information. Moreover, it confines the model's predictions within a very narrow scope and effectively reduces false positives in the segmentation.
>
> 2.	For your question about the prompt, bounding box prompts are derived from pseudo-labels generated by the ICL model, thus not necessarily more accurate than manually provided bounding box prompts. However, the advantage lies in eliminating the need for manual prompts for each query image, saving considerable human effort. The performance gains in our model are primarily relative to the ICL model (UniverSeg), with improvements mainly stemming from the semantic confidence maps generated by SAM and SAM's refinement of the output from the ICL model. Figure 4 demonstrates how ICL generates bounding box prompts and how SAM refines its outputs to some extent. Following your suggestion, we have included more examples in the appendix (Figure 5) to provide further clarity.
>
> Detailed Comments:
>
> 3.	For your inquiry regarding the coefficients mentioned in our paper, we have detailed the process of selecting our hyperparameters in Figure 6 of the appendix. In brief, we assessed the impact of various hyperparameters on fundus datasets and subsequently applied these hyperparameters to other datasets. We found them to be effective across datasets, thereby demonstrating the robustness and broad applicability of our algorithm.
>
> We extend our sincerest gratitude for your valuable suggestions, which have contributed to refining our manuscript.

---

### Official Review · Reviewer_N9RB · 2024-03-05

**Confidence:** 4
**Preliminary Rating:** 5
**Recommendation:** Oral
**Final Rating:** 5

**Summary:**

The authors propose to integrate the strong points of two growing fields in the segmentation domain: the robustness of the In-Context Learning approach to domain shift and the performance of the strong fundamental models such as SAM. They innovatively fuse these two approaches in a single framework, with a confidence map generation method to boost the performances of the whole. Extensive quantitative and qualitative studies on three modalities of data and five datasets prove the interest of the method.

**Strengths:**

The problem presentation is clear, and the proposed method is interesting and clearly explained as well. Notably, the figures are of a great help to support the textual explanations. Furthermore, the method has multiple innovative elements and an extensive validation.

**Weaknesses:**

- It would be interesting to consider at least another metric in addition to the Dice score, considering its properties/limitations (cf Reinke, Annika et al. “Common Limitations of Image Processing Metrics: A Picture Story.” ArXiv abs/2104.05642 (2021)), to get a clearer understanding of the behavior of the model.

**Detailed Comments:**

- Please precise which precise operations are done when you say in part 2.5 : "To eliminate noise within the prediction map, we first apply morphological shrinkage to the map, reducing it to 0.9 of its original size, followed by inflation to restore its size.". Are they differentiable?
- Which loss function did you use?
- Please precise what w and h stands for or how they're defined, and the quantity of data of the fundus datasets in the main body of the paper.

**Justification Of Final Rating:**

This paper is qualitative, being clear and easy to read, and it presents an innovative method that brings great improvement on multiple segmentation tasks on variable datasets. It has been studied in depth, with great effort to answer the reviewers' questions and remarks.

**Justification Of The Preliminary Rating:**

This paper is qualitative, being clear and easy to read, and it presents an innovative method that brings great improvement on multiple segmentation tasks on variable datasets. Very few critics to make.

**Questions To Address In The Rebuttal:**

Cf above

**Special Issue:**

Yes

---

> ### Author Response · Authors · 2024-03-15
>
> Thank you for your suggestions. I will respond to your points in order, from weaknesses to detailed comments:
>
> Weaknesses:
>
> 1.	For your suggestion to include additional metrics beyond the Dice score, we are actively working on incorporating extra metrics. However, due to the extensive number of experiments required in our study, approximately 3900 in total. For each task and each setting, we run 10 times. Given the constraints of our computational resources, we may not be able to complete this before the rebuttal deadline. We will update the results as soon as we finished.
>
> Detailed Comments:
>
> 2.	For your question regarding morphological operations, I would like to clarify that in our method, we have not fine-tuned the parameters of SAM (and MedSAM) and the ICL model (UniverSeg) on any specific dataset, meaning that we have not conducted backward propagation. This approach stems from the fact that SAM and UniverSeg have been extensively trained on a wide range of datasets, allowing us to utilize their parameters directly for segmentation across various medical domains. Consequently, the differentiability of morphological operations like shrinkage and inflation does not impact our work. To elaborate, for the pseudo-labels output by UniverSeg, we first apply morphological shrinkage to reduce the area of the pseudo-labels to 90% of their original size. This step helps eliminate some of the finer noise. Subsequently, we use morphological inflation to restore the retained pseudo-labels to their original size, ensuring the accuracy of the bounding boxes generated afterward. We clarify this in our revised manuscript.
>
> 3.	For your question about loss function, as mentioned in response to the previous query, we have not fine-tuned the parameters of the foundational models utilized in our study. Consequently, the application of a loss function was not a part of our methodology.
>
> 4.	Thank you for highlighting the need for clarification regarding the definitions of w and h. We apologize for the lack of clarity in our paper regarding this matter. Due to the varying spatial sizes associated with different layers in the network, it is necessary to continuously resize the semantic confidence map obtained from a specific layer for application to other layers. In this process, the spatial size of the target layer is denoted by h×w. We clarify this point in revisions of the paper. Regarding the quantity of data for the fundus datasets, this information is detailed in the main body of the paper as well.
>
> Your constructive feedback has been instrumental in enhancing the quality of our work, for which we are deeply thankful.

---

> > ### Comment · Reviewer_N9RB · 2024-03-21
> >
> > Thank you for your efforts in clarifying and completing this paper, including re-doing your experiments to complete your metrics, which I can understand is a tedious task. The answers you provided seems clear to me, so I'd like to thank you again for that. However, I understand that you do not fine-tune the foundational models used, thus having no need for differenciability in your morphological operations ; but don't you still need to train your Logistic Regression to generate your Semantic Confidence Map (and thus to use a loss function) ?

---

> > > ### Author Response · Authors · 2024-03-25
> > >
> > > Thank you for your understanding. We performed two-class classification using logistic regression with the binary cross-entropy as the loss function. This information has been added to Section 2.2 of our manuscript.
> > >
> > > Furthermore, we have incorporated a new metric—Average Symmetric Surface Distance (ASSD)—into Table 6. ASSD measures the average distance between corresponding points on the surfaces of two objects and is widely utilized in fundus and tumor segmentation. The results from ASSD further demonstrate that our method significantly improves UniverSeg, especially at small context sizes.

---

### Meta-Review · Area_Chair_1X5n · 2024-04-05

**Recommendation:** Accept (Poster)
**Confidence:** 4

**Metareview:**

This paper presents a method that combined in-context learning (e.g. universeg) for segmentation with interactive foundation models (e.g. SAM). Essentially the users use initial segmentations provided by universeg to then segment the target using SAM.

The reviewers all found the work intuitive and interesting, in an important and growing area. The discussion brought about some clarity for the reviewers, although none of them changed their scores. Overall, a solid paper that should be discussed at the conference.

In terms of presentation type, the paper received two votes for Poster (WA) and one for Oral (WR), and I do believe the decision on that is still borderline and either would be reasonable.

---

### Decision · Program_Chairs · 2024-04-06

Accept (Oral)